# Exploring the Impact of the Linker Length on Heat Transport in Metal–Organic Frameworks

**DOI:** 10.3390/nano12132142

**Published:** 2022-06-22

**Authors:** Sandro Wieser, Tomas Kamencek, Rochus Schmid, Natalia Bedoya-Martínez, Egbert Zojer

**Affiliations:** 1Institute of Solid State Physics, NAWI Graz, Graz University of Technology, 8010 Graz, Austria; sandro.wieser@student.tugraz.at (S.W.); tomas.kamencek@gmail.com (T.K.); 2Institute of Physical and Theoretical Chemistry, NAWI Graz, Graz University of Technology, 8010 Graz, Austria; 3Computational Materials Chemistry Group, Faculty of Chemistry and Biochemistry, Ruhr-University Bochum, 44801 Bochum, Germany; rochus.schmid@rub.de; 4Materials Center Leoben, 8700 Leoben, Austria; olganatalia.bedoya-martinez@mcl.at

**Keywords:** metal–organic frameworks, heat transport, thermal conductivity, structure-to-property, molecular dynamics, force field, NEMD

## Abstract

Metal–organic frameworks (MOFs) are a highly versatile group of porous materials suitable for a broad range of applications, which often crucially depend on the MOFs’ heat transport properties. Nevertheless, detailed relationships between the chemical structure of MOFs and their thermal conductivities are still largely missing. To lay the foundations for developing such relationships, we performed non-equilibrium molecular dynamics simulations to analyze heat transport in a selected set of materials. In particular, we focus on the impact of organic linkers, the inorganic nodes and the interfaces between them. To obtain reliable data, great care was taken to generate and thoroughly benchmark system-specific force fields building on ab-initio-based reference data. To systematically separate the different factors arising from the complex structures of MOF, we also studied a series of suitably designed model systems. Notably, besides the expected trend that longer linkers lead to a reduction in thermal conductivity due to an increase in porosity, they also cause an increase in the interface resistance between the different building blocks of the MOFs. This is relevant insofar as the interface resistance dominates the total thermal resistance of the MOF. Employing suitably designed model systems, it can be shown that this dominance of the interface resistance is not the consequence of the specific, potentially weak, chemical interactions between nodes and linkers. Rather, it is inherent to the framework structures of the MOFs. These findings improve our understanding of heat transport in MOFs and will help in tailoring the thermal conductivities of MOFs for specific applications.

## 1. Introduction

Metal–organic frameworks (MOFs) [1,2,3] are materials that have attracted substantial interest in the scientific community over the past two decades. They consist of inorganic nodes connected by organic linkers, forming highly porous structures. The multitude of possible configurations of the nodes and the diversity of potential linker structures lead to an unlimited number of conceivable materials with various degrees of porosity and a vast range of properties. In fact, there are already over 100,000 different MOF structures in the Cambridge Crystallographic Data Centre database [4,5]. Many potential applications of MOFs try to exploit their very high internal surface area or their adaptability through suitably arranging the building blocks. Some examples include gas storage [6,7], gas separation [8,9], catalysis [10,11,12,13], drug delivery [14,15], or electronic devices [16,17,18]. These applications often involve processes that generate heat, which needs to be dissipated. This can become a major bottleneck for large-scale implementations of MOFs, as they feature extremely low thermal conductivities [19,20,21]. For example, in the absence of heat dissipation, a significant increase in temperature (from 50 to 200 K) has been predicted for CO_2_ or CH_4_ adsorption in MOFs [19]. The use of MOFs for hydrogen storage in fuel cell applications [22,23,24,25] is also challenged by their low thermal conductivities, as efficient heat transport is crucial for maintaining effective operating conditions [26]. At the other end of the spectrum, the poor thermal conductivity of MOFs can be beneficial in ensuring effective thermoelectric energy conversion [27,28,29]. To design MOFs with the required properties, it is important to understand how structural modifications influence thermal conductivities.

Therefore, in recent years, several studies have been performed to advance our understanding of heat transport in MOFs. They revealed that MOFs often display an unusually small negative or even positive temperature dependence of the thermal conductivity [20,21,30,31,32] and they showed that defects such as missing linkers severely impede heat transport even at low concentrations [33]. Conversely, the interpenetration of frameworks results in materials whose thermal conductivities are essentially the sums of those of the sub-systems [34]. Additionally, the impact of phonon scattering rates on the MOFs’ thermal conductivities was studied for a variety of MOFs [35,36,37]. Changes in the phonon scattering behavior are also key to understanding variations in thermal conductivities due to different functional groups in the linkers of zeolitic imidazolate framework 8 (ZIF-8) systems [38]. In view of the above-mentioned applications of MOFs, several studies were also dedicated to understanding the impact of gas loading and the integration of guest molecules on heat transport, where, depending on the MOF and the loaded molecule, either a reduction [37,39,40,41] or an increase [31,40,42,43,44] in the thermal conductivity due to guest incorporation was observed.

Of conceptual interest for highly porous materials like MOFs is the impact of the porosity on thermal transport. Based on systematic modifications of simple model systems, it was suggested that the thermal conductivity is proportional to the inverse pore cross-sectional area [45]. A strong correlation between pore size and thermal conductivity was indeed found also for actual MOFs [19,46]. Different heat transport pathways due to different MOF topologies modify this simple picture [43,45]. The situation is further complicated by flexible MOFs, which undergo phase transitions that can change their pore sizes. A transition to more narrow pores like, for example, those found in MIL-53 has, indeed, lead to an increased thermal conductivity [46,47], while for systematically modified model systems it was shown that a change in the linker inclination angle reduces heat transport [48]. Interestingly, the strong impact of the pore size can lead to a situation where a disordered but denser phase of a MOF displays an increased thermal conductivity compared to its ordered counterpart [38,49]. Based on the available literature, it becomes clear that the structure-dependent heat transport in MOFs arises from a complex interplay of various factors.

Aiming at a better understanding of the specific mechanisms of heat transport in MOFs, we found in a previous investigation that the primary bottlenecks for thermal transport are the interfaces between the inorganic nodes and the organic linkers [50]. This suggests that increasing the length of the linker could lead to an improved thermal conductivity, as it decreases the density of interfaces in the heat-transport direction. However, longer linkers also mean a larger pore size, which, as mentioned above, is detrimental to thermal transport due to a reduced number of possible heat transport channels [19,43,44,45,51]. In the present study, we analyzed the interplay between these effects employing non-equilibrium molecular dynamics (NEMD) simulations for a series of isoreticular MOFs (IRMOFs). Starting from the structure of IRMOF-1, we systematically increased the number of phenylene units in the linkers in order to increase their lengths. Moreover, we analyzed the effect of a rigidification of the linker backbones and studied the impact of structural anisotropy by increasing the linker length only in the heat-transport direction.

Besides studying actual MOF structures, we also investigated a series of model structures, for which variations in linker lengths become straightforward without simultaneously changing other relevant structural details of the system. Consequently, these model structures are useful for separating geometry and symmetry-related effects from the impact of the chemical details of the actual MOF structures.

On more technical grounds, it is imperative to employ appropriately parametrized and rigorously benchmarked force fields in the NEMD simulations in order to make meaningful predictions. Thus, for all of the studied systems, we also thoroughly assessed the impact of our force-field parametrization by comparing force field and ab initio calculated properties of phonons and other heat-transport-related parameters.

## 2. Studied Systems

The reference structure used in this work is the isoreticular metal–organic framework 1 (IRMOF-1), also known as MOF-5 [1]. It is built from zinc-oxide nodes connected by 1,4-benzenedicarboxylate (BDC) linkers forming a cubic framework (Figure 1a). Combining the basic framework of IRMOF-1 with the organic linkers shown in Figure 1b-e (before deprotonation) yields the MOFs studied here. IRMOF-10 and IRMOF-16 are realized by including a second and third phenylene unit into the linker backbone. As an example for a rigidified and laterally extended system, a pyrene-based linker was used to build IRMOF-14. This system has almost the same linker length as IRMOF-10, but has a significantly different linker mass and a reduced possibility for the linker to undergo torsional vibrations. All systems mentioned so far have been successfully synthesized before [52,53]. In order to evaluate the impact of expanding only the linker in the heat-transport direction while maintaining the IRMOF-1 structure perpendicular to it, we also studied a hypothetical, anisotropic system in which the framework is connected by BDC linkers in two Cartesian directions and a pyrene linker in the third direction. This material will be referred to as MOF-1-1-14.

As mentioned above, we also studied model systems mimicking the heat-transport pathways in MOFs with linear linkers and octahedral nodes (see Figure 1f). These model MOFs are built from identical “virtual atoms” linked via bonded force field terms. The nodes consist of octahedral arrangements of these “virtual atoms”. Within the nodes, the main heat transport pathways extend from the carboxylate group of one linker via the neighboring Zn atoms, over the carboxylate groups of perpendicular linkers and the next group of Zn atoms, to the carboxylate group of the linker opposite the first one (see Figure 1g). The central oxygen atom in each node is not explicitly considered in this model, as one can show that its interaction with the neighboring atoms has only a very minor impact on the heat transport through the node. This is discussed in detail in the Appendix A. The number of virtual atoms in the linkers is systematically varied either in all directions or only in the direction of heat transport. The masses of the virtual atoms and the interaction strengths are intentionally kept identical throughout the entire model system in order not to obscure geometry-related trends by “chemical details”. The impact of such chemical details has already been addressed for IRMOF-1 in Ref. [50] by varying the metal atoms in the nodes (in this way changing node masses and node-linker bonding strengths). Additional details motivating the choice of the model systems, the parameters used in the model, and their impact on the most relevant investigated quantities, are described in the Appendix A.

The three-dimensional visualizations of atomistic structures in this work were created using the VESTA software (Visualization for Electronic and STructural Analysis, 3.5.7, 2021, Koichi Momma and Fujio Izumi, Tokyo, Japan) [54].

## 3. Employed Methodology

In the following section, the procedure for obtaining thermal conductivities by means of molecular dynamics (MD) simulations using suitable force fields will be detailed. Additionally, we will introduce the quantities of interest investigated in this work.

### 3.1. Simulation Procedure

The MD simulations were carried out using the Large-scale Atomic/Molecular Massively Parallel Simulator (LAMMPS, 2 July 2021, Sandia National Labs and Temple University, NM, USA) [55] and by employing 3D periodic boundary conditions. In the following, we provide a short overview of the approach. Further details can be found in the Appendix A. In all MD simulations, a time step of 0.5 fs was used. The equilibrium volume was calculated at zero pressure and 300 K for each system in an isothermal-isobaric ensemble (NPT). This cell volume was used to compute the thermal conductivity using non-equilibrium molecular dynamics (NEMD) simulations (as schematically depicted in Figure 2) [56]. For this, the atoms were first equilibrated to a temperature of 300 K within an isothermal-isochoric ensemble (NVT). Next, two slabs separated by one-half of the simulation cell length were designated as heat source and heat sink. The temperature in these regions was set by two independent Langevin thermostats to temperatures of (300 ± 50) K (the exact temperature difference did not impact the result beyond the reported error; see the Appendix A for further details). Once the steady-state was reached, the resulting heat flux J from the hot to the cold thermostat was determined from the total energy added or subtracted by the Langevin thermostats. The temperature gradient ∇T was obtained from the total temperature difference between the thermostats divided by their separation distance, as it was recently shown [57] that the region comprising the nonlinear temperature jump at the thermostat boundary should not be disregarded (as is often done in NEMD simulations). Both J and ∇T were averaged for a total simulation time of 10 ns after the steady state had been reached. This allowed the calculation of the thermal conductivity κ using Fourier’s law:(1)J=−κ∇T

The value of κ obtained from this approach is still prone to finite size effects due to the relatively small size of the simulation box, which leads to scattering effects at the thermostat boundaries [56,57]. In order to provide proper thermal transport properties for extended crystals, several NEMD simulations with different cell lengths *L* in the heat transport direction were performed. The infinite size limit was then obtained via a 1/*L* extrapolation of the obtained 1/*κ* values [56] (for further details, see Appendix A). For the actual MOFs, supercells containing 8 to 24 cubic unit cells in the heat transport direction were used. This corresponds to cell lengths of 200–1000 Å. Perpendicular to the heat transport direction, we observed convergence for cross sections consisting of 2 × 2 cubic unit cells, which is larger than the 1 × 1 cells employed in [50]; convergence issues in these earlier simulations are discussed in the Appendix A. Similar cell sizes were used for the model systems, ensuring convergence with the simulation cell length parallel and perpendicular to the heat transport direction (see Appendix A).

### 3.2. Parametrization of the Force Fields

Classical force fields (FFs) were used to describe interatomic interactions. Several traditional force fields exist, which can model bond lengths and the general structure of even complex materials like MOFs rather well. These comprise, for example, the universal force field (UFF) [58] or the Dreiding [59] force field. They are, however, known to have problems in providing an accurate description of vibrational properties [60], which are a crucial prerequisite for the calculation of heat transport properties. Therefore, in order to assess to what extent “off-the-shelf” transferrable force fields are suitable for calculating thermal conductivities of MOFs, we employed the Dreiding force field and a variant of the UFF, referred to as UFF4MOF [61,62], in order to model heat transport in IRMOF-1. UFF4MOF is an expanded version of the universal force field designed to model coordination environments common in metal–organic frameworks.

Considering that the tests employing the above off-the-shelf FFs were by no means satisfactory (see Results and Discussion section), we eventually resorted to non-transferrable FFs, which were parametrized system-specifically based on periodic ab-initio reference geometries and Hessian matrices. These data were computed in the framework of density functional theory (DFT), employing the Vienna ab-initio Software Package (VASP, 5.4.4, 2017, VASP Software GmbH, Vienna, Austria) [63,64,65,66,67,68], relying on the Perdew–Burke–Ernzerhof (PBE) functional [69,70], and applying the D3 dispersion correction with Becke–Johnson damping [71,72]. The functional form of the FFs is derived from MOF-FF [73], a force field specifically developed for MOFs. Compared to the traditional MOF-FF approach, the primary modification in the present work is the inclusion of additional terms describing the coupling of bond lengths between the next-nearest neighbors. Other popular force fields, like COMPASS [74], have successfully employed such terms to describe the situation in organic molecules. Here, we find them to be beneficial for modelling the vibrational properties of the studied MOFs.

The FF parametrizations were performed by employing the ff_gen software (2021, Ruhr University Bochum, Bochum, Germany), which has also previously been used to obtain MOF-FF potentials [75]. Great care was taken to choose proper weights, starting values, and limits for individual interaction parameters. These limits improve convergence and prevent unphysical interactions arising from the high-dimensional fitting procedure. This typically yielded a satisfactory description of phonon band structures and atomic forces in randomly displaced structures (see Section 4.1 and Appendix A in the Appendix A). The only exception was IRMOF-10, for which the above-described FF parametrization produced rather poor results. We attribute this to the observation that for this linker, the torsion angles between the rings and the carboxylates for symmetry reasons cannot adopt the optimum values that they would have in the isolated molecule. Thus, in order to obtain a satisfactory performance of the FF also for IRMOF-10, we included forces from 100 random off-equilibrium structures as additional references in the parametrization process.

Beyond the FF parametrization, the necessary atomic charges were computed from the electrostatic potential of the DFT-based reference data using the Repeating Electrostatic Potential Extracted ATomic (REPEAT) methodology [76]. In order to calculate force constants (Hessian matrices), phonon band structures and phonon group velocities, the phonopy [77] package was used in combination with VASP and LAMMPS. For more details concerning the parametrization strategy for the FF of each system and the benchmarking against ab initio data, see the Appendix A Appendix A. This thorough benchmarking was particularly important, as each of the potentials for the different systems not only comprises certain conceptual differences in the interatomic interactions, but also includes hundreds of variables. This required different setups of initial guesses and weights due to differences in the convergence behavior.

### 3.3. Quantities of Interest

The thermal conductivity κ is not the only quantity of interest in this study, as we intend to separate the contributions of nodes, linkers, and interfaces. In this context, a complication arises because isotropically increasing the linker length simultaneously decreases the linker densities. Thus, in the following, several quantities need to be defined that account for these aspects.

From the temperature profiles of the NEMD simulations in Figure 2, it becomes apparent that there is a strong correlation between the atomistic structure of the material and the local effective temperature gradient. Three different components with their thermal conductances and resistances are identified: the organic linker, the inorganic node, and the interface between linker and node (in the following indexed by the subscript i) [50]. The primary quantities of interest are the heat transfer coefficients, $s_{A,i},$ which are the thermal conductances, $S_{i}$, per area, A, and the thermal resistances, $R_{i}=1/S_{i}$. When multiplied by the area, they yield the thermal insulances, $r_{A,i}$.
(2a)$$s_{A,i}=\frac{S_{i}}{A}=\frac{1}{R_{i}A}=\frac{1}{r_{A,i}}$$

The link between the thermal conductance/resistance and the thermal conductivity intrinsic to a specific component of the MOF follows from simple geometrical arguments as:(2b)$$S_{i}=\frac{1}{R_{i}}=\kappa_{i}\frac{\Delta z_{i}}{A},$$
where $\Delta z_{i}$ is the length of the object of interest (in the heat-transport direction). The heat transfer coefficients are intuitively more easily accessible, but the insulances have the advantage that they are cumulative, i.e., the total thermal insulance of a complex structure can simply be calculated from the sum of the insulances of its components. As mentioned above, a complication in this context is that increasing the linker length in an isoreticular MOF also decreases the number of heat transport channels per area. Thus, it is useful to additionally define the equivalent quantities per linker. Then, $s_{N,i}$ represents the thermal conductance per heat transport channel and $r_{N,i}$ is its inverse, which we will refer to as the thermal transport-channel insulance.
(3)$$s_{N,i}=\frac{S_{i}}{N}=s_{A,i}\frac{A}{N}=\frac{1}{R_{i}N}=\frac{1}{r_{A,i}}\frac{A}{N}=\frac{1}{r_{N,i}}$$

Here, N/A denotes the number of linkers per area parallel to the heat-transport direction. Inserting Equations (1) and (2) into (3) yields a relation between $r_{N,i}$ (or $r_{A,i}$), the heat flux J, the temperature gradient ∇T, the number of linkers per area, and the thermal conductivity.
(4a)$$r_{A,i}=-\frac{\nabla T}{J}\Delta z_{i}=\frac{\;1}{\kappa_{i}}\Delta z_{i}.$$
(4b)$$r_{N,i}=-\frac{\nabla T}{J}\frac{N}{A}\Delta z_{i}=\frac{\;1}{\kappa_{i}}\frac{N}{A}\Delta z_{i}.$$

Equations (4) show that the thermal transport-channel insulance is inversely proportional to the thermal conductivity and is linearly proportional to the number of linkers/nodes per area and the spatial extent of the linkers/nodes. The linear scaling with linker density can be understood from the fact that for a constant thermal insulance of each linker, increasing the linker density results in an increase in the thermal conductivity. While Equation (4) is suitable for thermal resistances of spatially extended moieties (like linkers and nodes), for the node-linker interface it is useful to define an analogous quantity, which accounts for the fact that the interface has no spatial extent. This can be done in analogy to the Kapitza resistance, $R_{Kapitza}$, which is proportional to the temperature drop, δT, at the interface:(5)$$r_{N,interface}=r_{A,interface}\;\frac{N}{A}=R_{Kapitza}\frac{N}{A}=-\frac{\delta T}{J}\frac{N}{A}.$$

As indicated above, the total thermal insulance of the “thermal repeat unit” consisting of one node, one linker, and two node-linker interfaces is simply given by the sum of the respective contributions.
(6a)$$r_{A,unit}=r_{A,linker}+r_{A,node}+2r_{A,interface}$$
(6b)$$r_{N,unit}=r_{N,linker}+r_{N,node}+2r_{N,interface}$$

These quantities, which will be central to the following discussion, can then be linked to the thermal conductivity of the MOF:(7a)$$\kappa =\frac{\Delta z_{unit}}{r_{A,linker}+r_{A,node}+2r_{A,interface}}$$
(7b)$$\kappa =\frac{\Delta z_{unit}}{r_{N,linker}+r_{N,node}+2r_{N,interface}}\times \frac{N}{A}.$$

Here, $\Delta z_{unit}$ is the length of the thermal repeat unit in the heat transport direction and comprises one linker, one node and two interfaces. Strictly speaking, with this choice of thermal repeat unit, $\Delta z_{unit}$ corresponds to only half of the length of the conventional cubic unit cell of the investigated MOFs (see Appendix A). The reason for this is that two adjacent linkers are always twisted in opposite directions, which leads to a doubling of the crystallographic unit-cell length. The different twists, however, have no significant impact on thermal transport such that the resulting differences in $r_{A,i}$ and $r_{N,i}$ between neighboring linker-node sections are within the noise level and are, therefore, not considered separately.

To calculate the relevant quantities from the heat flux, the temperature gradients, and geometric parameters, we employed the following strategy and conventions: the interface between node and linker was defined to be exactly halfway between the terminal O atoms of the linker and the Zn atoms of the node. The temperature profiles used for the evaluation were obtained from the time-averaged kinetic energies (${\bar{T}}^{'}=2/\left(3k_{B}\right)\cdot {\bar{E}}_{kin}$) of the individual atoms, which we refer to as the “local effective temperatures”. Additionally, only components with a distance of at least 50 Å from the thermostat were used for the evaluation of the transport-channel insulance to avoid inconsistent contributions from the nonlinear region close to the thermostat boundary. The temperature profiles of nodes and their neighboring linkers were then used to perform linear fits from which node and linker resistance contributions were extracted, employing Equation (4). The remaining temperature step was associated with the interface resistance using Equation (5). To remain consistent with the determination of the total thermal conductivity, for which the regions close to the thermostat were also considered (see above), the individual transport-channel insulance contributions were rescaled accordingly (for more details see the Appendix A). Finally, it should be mentioned that, similar to the situation for the thermal conductivity, the evaluated transport-channel insulance contributions are affected by finite-size effects. These were corrected employing an approach equivalent to that described above for determining κ (for more details, see the Appendix A Appendix A).

## 4. Results and Discussion

### 4.1. Assessing the Employed Force Fields

As a first aspect, the results for conventional, transferrable force fields shall be discussed briefly: calculations on IRMOF-1 employing the UFF4MOF and the Dreiding force fields at 300 K predict thermal conductivities of 0.847 W/(mK) and 1.102 W/(mK), respectively. Both values are significantly higher than the experimental room temperature value of 0.32 W/(mK) [21]. We primarily attribute this to the much higher phonon group velocities obtained with these force fields compared to the ab initio DFT simulations (see Appendix A in the Appendix A). Additionally, the force fields mentioned above describe the bonding interactions between atoms by harmonic potentials. This is clearly a suboptimal choice when modeling heat transport, which is crucially impacted by phonon lifetimes as an intrinsically anharmonic property. As an alternative, one could employ more sophisticated force fields designed for use with MOFs, like BTW-FF [78] or the original (partly transferrable) version of MOF-FF [73], with common parameters for specific segments. As detailed in Section 3.2, we here opted for a fully system-specific variant of MOF-FF, for which all parameters were determined from scratch for each MOF (as outlined in Section 3.2 and in the Appendix A in Appendix A). The expected higher level of accuracy is particularly important here, bearing in mind that we study relatively small differences in heat transport triggered by rather subtle structural variations of a specific subset of MOFs.

To ensure the reliability of the obtained FFs, we benchmarked them against various observables, either from experiments or from ab initio simulations. The first set of observables comprises the lattice parameters of the optimized crystal structures, for which experimental reference data exists (see Table 1). Here, the performance of the FFs is highly satisfactory, both at the hypothetical temperature of 0 K, where the FF results are compared to the DFT simulations, as well as at a temperature of 250 K, where FF-based molecular dynamic simulations employing an NPT ensemble are compared to experimental data. Another observation that concerns the anharmonicity of the potential energy surface is that the FF-based simulations suggest a negative thermal expansion for all investigated materials. This is consistent with the experimental observation of such a negative thermal expansion, which is a rather common peculiarity in many MOFs [79,80,81]; for example, for IRMOF-1 the simulated thermal expansion coefficient amounts to −15.7 × 10^−6^ K^−1^ (obtained from a linear fit of lattice parameters in a temperature range between 100 K and 400 K, as detailed in the Appendix A in Appendix A). This is in excellent agreement with experimentally measured values ranging between −13.1 × 10^−6^ K^−1^ and −15.3 × 10^−6^ K^−1^ [82,83] (determined at 300 K). A similarly good agreement is found for the geometric parameters describing the atomistic structure of the MOF, such as bond-lengths, bond angles, and torsions (see Appendix A).

In addition to geometrical parameters, vibrational properties and the associated force constant are of particular relevance for benchmarking the force fields for thermal transport studies. In this context, Figure 3a shows for the prototypical case of IRMOF-1 that the vibrational frequencies at Γ (the center of the first Brillouin zone) in the entire spectral region agree very well between FF and DFT calculations. This agreement is still good when zooming into the low-frequency region, which comprises the phonons that are most important for thermal transport (see Figure 3b) [36]. A similarly good performance of the force fields is observed for the other MOFs, as shown in Appendix A in the Appendix A. The same applies to the Hessian matrix, for which a comparison between FF and DFT data is contained in Appendix A.

In order to quantify the similarity of the vibrational displacement vectors, we calculated the dot-products of the associated eigenvectors of the Γ-phonons obtained in the DFT and MD simulations. A histogram for the values of these dot products for IRMOF-1 is contained in Figure 3c. This again testifies to the excellent correspondence of FF and DFT results. Notably, an even better agreement is found for MOF-1-1-14, but it should also be mentioned that the dot products for IRMOF-10 and IRMOF-16 are lower on average. Nevertheless, as shown in Appendix A, in these systems the dot products for nearly all displacement vectors are >0.5 as well.

Finally, to not only analyze the accuracy of the FFs at the energetic minimum, we additionally tested the agreement of the Cartesian atomic forces on the atoms for sets of random displacements within the unit cell. Each atom in the primitive unit cell of the respective systems was displaced by a random value given by a normal distribution with a standard deviation of 0.01, 0.05 or 0.10 Å. For each distance, ten different displacements were generated and calculated. As shown for IRMOF-1 in Figure 3d, good agreement between the FF and DFT results is obtained, especially for moderate forces. Similar behavior is observed for the other MOFs (see Appendix A).

The above considerations show that for the MOFs in the focus of the present study we have parametrized a set of system-specific force field potentials that ensure a good description of several of the main ingredients of thermal transport (in particular structural and vibrational properties). This should allow for reliable NEMD simulations as the basis for the following analysis. Indeed, as far as the thermal conductivity is concerned, for IRMOF-1 (to the best of our knowledge, the only considered material for which corresponding experimental data are available), the calculated value of 0.29 W/(mK) for the thermal conductivity agrees very well with the experimental value at room temperature, amounting to 0.32 W/(mK) [21]. In passing, we note that the value of 0.41 W/(mK) we reported for IRMOF-1 in a previous study [50] is less reliable, for reasons discussed in detail in Appendix A.

### 4.2. Trends in Thermal Conductivities

Before analyzing the calculated thermal conductivities, it is useful to discuss the trends one would expect from Equation (7b) based on certain approximations: for that we assume that $r_{N,node}$ and $r_{N,interface}$ are independent of the nature of the linker, while $r_{N,linker}$ scales linearly with the linker length. This is in analogy to the expectations for classical electrical or thermal resistors and builds on the notion that extending the linker without changing the docking chemistry would neither affect the thermal resistance of the nodes nor that of the interfaces. A comparison between the expected trends and the actual results of the NEMD simulations will then reveal to what extent these assumptions are justified. For this, we set $r_{N,node}$ and $r_{N,interface}$ to the values calculated for IRMOF-1 (see below) and the linear scaling for $r_{N,linker}$ is also chosen such that, for an extent of the thermal repeat unit like in IRMOF-1, the $r_{N,linker}$ value of IRMOF-1 is reproduced.

When increasing the length of the linker only in the heat transport direction, under these assumptions the contribution of $\Delta z_{unit}$ in the numerator of Equation (7b) dominates, as the only other linker-length dependent quantity (the linker transport-channel insulance, $r_{N,linker}$) is comparably small. The consequence is an almost linear increase in the thermal conductivity with the linker length (see dashed black line in Figure 4). The reason for this trend is that extending the linker length in the heat transport direction primarily results in a decrease in the density of node-linker interfaces acting as heat transport bottlenecks. Conversely, when increasing the length of linkers in all directions, the impact of the increase in pore size dominates. In this case, the area A in Equation (7b) scales quadratically with $\Delta z_{unit}$, such that, as an overall trend, κ should drop approximately with $1/\Delta z_{unit}$ (with an additional weak impact of the length-dependence of $r_{N,linker}$). The overall trend is shown as a solid black line in Figure 4.

Finally, when completely disregarding the details of the atomistic structure of the MOFs, the only impact of the linker length in an isoreticular MOF should be a decrease in the thermal conductivity due to the increasing unit-cell and pore cross-section. In this case, one would expect a drop of κ with $1/{\left(\Delta z_{unit}\right)}^{2}$, as then only the impact of A in Equation (7b) prevails. This situation is depicted in Figure 4 as a dotted blue line. In fact, such an evolution has previously been suggested in the literature as a consequence of increasing the linker lengths based on model MOFs [45] (for a visualization of the thermal conductivity values as a function of the inverse pore cross-section, see Appendix A in the Appendix A).

A comparison of the actual NEMD results with the expectations based on the simple models from above yields a few surprises (see Figure 4): For the series comprising IRMOF-1, IRMOF-10, and IRMOF-16, the drop in κ with the linker length is much more pronounced than one would expect upon extending the linkers in all directions (solid black line). In fact, the corresponding data points are essentially aligned along the line depicting the trend obtained when only considering the increase in pore size, disregarding the impact of a decreased density of heat-transport bottlenecks. This is possible only if, in contrast to the model assumption, the thermal transport-channel insulance of the interface does depend on the linker length.

The situation changes fundamentally for IRMOF-14, comprising pyrene-based linkers: for this system, the linker length is very similar to that of IRMOF-10, but its thermal conductivity lies clearly above the expectation for extending linkers in all directions. Interestingly, when employing pyrene-based linkers only in the direction of the heat flux and BDC linkers perpendicular to it (i.e., in MOF-1-1-14), an intermediate situation is obtained: The actual NEMD calculated thermal conductivity of MOF-1-1-14 lies below the trend for extending linkers only in the heat-transport direction (dashed black line in Figure 4). However, it is clearly higher than the thermal conductivity of IRMOF-1, which is in line with the qualitative expectation derived from a decreased density of interfaces. This implies also that the linkers perpendicular to the direction of the heat flux have a distinct impact on the thermal conductivity beyond merely changing the density of heat-transport channels. This observation is not entirely unexpected considering that the perpendicular linkers do have a distinct effect for the actual phonon spectrum of the MOF. These results call for an in-depth analysis of the individual contributions to the thermal insulance in the different MOFs.

### 4.3. Thermal Resistance Contributions

In order to provide such an analysis, it is useful to first consider the actual local effective temperature profiles in the NEMD simulations, as they form the basis for the distinction between node-, linker-, and interface contributions [50]. Figure 5a shows the average temperature profiles for all nodes and adjacent linkers obtained in NEMD simulations of approximately equally long supercells for each system. Step-like temperature profiles can be seen for all systems with low-temperature gradients across linkers and nodes and an abrupt temperature step at the interface. This is analogous to our findings for IRMOF-1 as discussed in Ref. [50]. In general, all profiles look relatively similar, as the primary difference for the thermal conductivities in Figure 4 results from the differences in the heat flux between heat source and the heat sink for a given temperature gradient.

Analyzing the situation in more detail, one sees that the central parts of the linkers display extremely low thermal resistances (as can be inferred from the diminishing temperature gradient). The efficient heat transport across the linker is consistent with previous observations of excellent heat transport properties of extended π-conjugated organic materials [86]. Conversely, there is a substantially more pronounced temperature drop in the region of the carboxylate group. A noticeable aspect is that, for longer linkers, the temperatures of the O and C atoms in the carboxylate group become more similar, which results in a widening of the temperature gap to the rest of the linker. This trend coincides with an increase in the force constants between the C and O atoms observed for longer linkers (for specific details, see the Supplementary Material Appendix A). The temperature step between C and O could be interpreted as the formation of a second interface, acting as yet another heat-transport bottleneck. However, in the following section, we will not include such a second interface in the analysis of the thermal insulances of the individual components, as (i) the magnitude of the temperature step between the carboxylic carbon and its neighbor is much smaller than that of the step between the carboxylic oxygens and the neighboring Zn atoms; and (ii) the temperature profiles are too noisy to extract reliable insulances for the second interface. As a consequence, its impact will be included in the thermal insulances of the linkers and, thus, will be the primary origin of the linker-length dependence of $r_{N,linker}$.

Figure 5b shows the calculated thermal transport-channel insulance contributions for all of the investigated MOFs. As discussed in detail in Section 3.3, they no longer contain the impact from variations of the cross-sectional pore size and are, thus, representative of the properties of individual linkers. In this sense, thermal transport-channel insulances can be used to analyze the deviations of the actual thermal conductivities from the trends depicted by the solid and dashed black lines in Figure 4.

As expected from the temperature profiles, the interface contributions dominate in all systems. Interestingly, the $r_{N,interface}$ values vary significantly between the different MOFs, as inferred already from the linker-length dependence of the thermal conductivities. In fact, $r_{N,interface}$ essentially doubles from IRMOF-1 to IRMOF-16. As will be detailed in Section 4.4, a pronounced (roughly linear) increase in the interface transport-channel insulance can, actually, be attributed to the framework structure and topology of the investigated class of MOFs. The value of $r_{N,interface}$ for IRMOF-10, however, strongly deviates from such a linear trend, as it is essentially as large as the value for IRMOF-16 despite the clearly shorter linker. This particularly high contact insulance of IRMOF-10 can, actually be associated with fundamental differences in the atomistic structure of the MOF: Ideally (i.e., for an isolated linker molecule), the phenylenes and the adjacent carboxylic acids would be coplanar, while for isolated biphenyl molecules, the ideal twist angle between phenylene units amounts to around 40° (44.4° in biphenyl [87]). Moreover, in the type of MOFs studied here, the two carboxylate groups of each linker and the central parts of the nodes would ideally also lie in the same plane. All these conditions can be simultaneously fulfilled in IRMOF-1, IRMOF-14, MOF-1-1-14, and largely also in IRMOF-16. It is, however, not possible in IRMOF-10 due to the structure of the linker, as the twist between the two phenylenes prevents the carboxylates from being in the same plane. This results in deviations from the ideal twists (see Figure 5c). An equivalent situation is expected for all linkers with an even number of twisted repeat units. Consequently, the entire MOF structure becomes strained. As shown in Figure 5c (bottom panel), the resulting distortion affects not only the linkers but also the nodes. The data in Figure 5b suggest that in this strained structure phonon scattering is increased such that the thermal insulances of the interface and of the node rise. The strained structure of IRMOF-10 also results in a reduction of the Zn-O force constants (by about 5–10% compared to the other systems; see Appendix A Appendix A), which is another factor increasing $r_{N,interface}$ [50].

For the pyrene-linked IRMOF-14, the interface resistance is essentially the same as for IRMOF-1 despite the longer linkers. This is the main reason for the rather large thermal conductivity of the system. Compared to the equally long IRMOF-10, one factor explaining the improved performance of IRMOF-14 is that the rigid backbone prevents the strain discussed in the previous paragraph. Notably, in IRMOF-14, $r_{N,interface}$ is also smaller than what one would expect based on the results for the model systems discussed below (in Section 4.4). For those, a longer linker length leads to a larger interface transport-channel insulance. To what extent this is a consequence of the suppression of torsional vibrations or the large moment of inertia of the pyrene is not clear at this stage.

An aspect that is confirmed by a comparison between IRMOF-14 and MOF-1-1-14 is that the cross-sectional linkers perpendicular to the heat flux direction have a profound impact on the coupling between nodes and linkers, with the heavier and more extended perpendicular linkers in IRMOF-14 apparently posing a clear advantage for transport through each heat-transport channel.

The thermal insulances of the linkers represent the smallest contribution to the overall thermal resistance in all studied systems, even though they also contain the impact of the rather steep drops in the effective local temperature in the region of the carboxylates (see above). Still, it is worthwhile discussing a few observations: compared to IRMOF-1, one observes a distinct reduction of $r_{N,linker}$ for the more extended and rigid pyrene-based linkers in IRMOF-14 and MOF-1-1-14. This is consistent with previous findings for polymers, that less flexible organic backbones are beneficial for heat transport [86]. Not unexpectedly, the largest value of $r_{N,linker}$ is obtained for IRMOF-10, which can again be attributed to the strained nature of that MOF. The latter also explains why the thermal transport-channel insulance of the nodes is the largest in IRMOF-10.

A commonly applied approach for analyzing heat transport through interfaces relies on a comparison of the densities of phonon states for the materials at either side of the interface [88,89]. In such a comparison, a larger overlap of the densities of states at an interface is interpreted as a strong similarity of the phonons in the two sub-systems, which leads to a reduced phonon scattering. In our case, we would have to consider the node and linker as separate materials, but they are both part of the same system. This is why we compared the densities of phonon states projected on the different components instead. Unfortunately, we were not able to extract insights from such an analysis that could explain the observed differences in interface resistances (for specific details, see the Appendix A). This is most likely due to the large number of “inactive” phonon modes in the low-frequency region, which do not contribute to heat transport due to their low group velocities, but which are still part of the density of states.

To generate additional insights regarding the origin of the observed trends, we therefore designed model systems, which allow more systematic modifications of the linker lengths without simultaneously changing the majority of the other system parameters (like the degree of strain in nodes and linkers and variations in bonding strengths).

### 4.4. Analyzing Instructive Model Systems

The general setup of the model systems consisting of “virtual atoms” with equal masses and interaction strengths has already been described in Section 2. Their general structure is shown in Figure 1f. Also for the model systems, we changed the lengths of the linkers either isotropically (i.e., in all directions) or anisotropically (i.e., only in the heat transport direction). For the latter case, the linkers perpendicular to the heat transport direction contained five “virtual atoms”.

Again, for each model system, NEMD simulations were performed at a temperature of 300 K. Figure 6a shows the averaged temperature profiles from simulations of approximately equally long supercells, when the linkers are extended in all directions. The data for the anisotropic systems are qualitatively very similar and can be found in the Appendix A in Appendix A. The temperature profiles in Figure 6a look conceptually strikingly similar to those of the real-MOF counterparts in Figure 5a. Despite the absence of any chemical “details” (like heavy and light atoms, weak and strong bonds, etc. …), there is a very low temperature gradient in the nominal linker region, a somewhat larger gradient in the nominal node region, and a massive temperature drop at the interface between the two. The latter again represents the primary heat transport bottleneck. The observation that this occurs for structures consisting of identical “virtual atoms” shows that the existence of the heat-transport bottleneck is not triggered by a particularly high difference in the masses of linkers and nodes or by a particularly weak coupling between metal ions and carboxylic oxygens. It is, apparently, rather a direct consequence of the geometric setup of the framework. Nevertheless, it should be mentioned that differences in masses and metal–oxygen couplings do impact the actual magnitude of the temperature step and, thus, can be used to tune the thermal conductivity of such structures, as discussed in Ref. [50].

Considering the conceptual similarity between the model systems and the actual MOFs, it is worth plotting the dependence of the (finite-size corrected) thermal conductivity as a function of the linker length. The results for isotropic and anisotropic model systems are shown in Figure 6b. Moreover, the expected trends for constant node and interface thermal insulances and linearly length-dependent linker insulances derived from Equation (7b) are shown as a solid (dashed) black line. The dotted blue line again depicts the situation in which only the decreased density of heat-transport channels due to an increasing pore size is considered. For the lines in Figure 6b, the cubic model system with five “virtual atoms” as linkers serves as a reference. Interestingly also for the evolution of κ, the trends for the model system are strongly reminiscent of the situation in the actual MOFs. In particular, the thermal conductivities of the isotropic systems almost directly follow the purely “geometric” trend (i.e., the dotted blue line) and the anisotropic systems display only a comparably weak increase of κ with the linker length.

The reason for this behavior can be inferred from the plot of the thermal transport-channel insulances in Figure 6c: as for the actual MOFs, the contribution of the interface dominates for nearly all linker lengths, while $r_{N,linker}$ consistently has the smallest value. Especially for the isotropic system, $r_{N,interface}$ increases steeply (essentially linearly) with the linker length. This reveals the intrinsic dependence of the interface transport-channel insulance on the linker length without additional interference from aspects like strained linkers and nodes or other aspects of the chemical “fine structure” of the MOF. The data in Figure 6c also show that this close to linear increase of $r_{N,interface}$ prevails for linkers with comparably many repeat units (many more than one could test for actual MOFs). Conversely, $r_{N,node}$ and especially $r_{N,linker}$ in Figure 6c display a much less pronounced dependence on the linker length, again in analogy to the actual MOFs. A comparison between the anisotropic and the isotropic model systems reveals that in the anisotropic case, one encounters a much more gradual increase of $r_{N,interface}$. This again confirms the notion that the increase in the interface insulance is also directly connected to the properties of the linkers perpendicular to the heat–transport direction.

These results clearly illustrate the reason for the failure of the original hypothesis that increasing the linker length would improve heat transport at least per heat-transport channel/linker: the deviation from the expectations is apparently not rooted in the chemical details of the studied MOFs, but it is rather a direct consequence of the network structure and topology. Nevertheless, carefully tuning the linker structure can apparently help to diminish that problem, as is shown by the data for IRMOF-14 with similar interface insulances as IRMOF-1.

## 5. Conclusions

We systematically studied the impact of the linker length on heat-transport in MOFs by means of non-equilibrium molecular-dynamics simulations. Based on previous findings that the node-linker interfaces represent the bottlenecks for heat transport, the expectation was that extending the linkers should decrease the density of these bottlenecks and, thus, should be highly beneficial for thermal transport. On a more macroscopic level, it should at least in part compensate for the effect of the increase in pore size associated with longer linkers.

On more technical grounds, comparing thermal transport in different MOFs is complicated because variations in the thermal conductivities between different systems can be relatively small. Thus, great care was taken to consistently parametrize high-level system-specific force fields against state-of-the-art ab initio data and to benchmark them thoroughly against a variety of relevant theoretical and experimental data.

Interestingly, in the simulations, the above expectations considering a beneficial impact of decreasing the interface density by extending the linkers were not met for IRMOFs comprised of (oligo)phenylenedicarboxalytes, linking zinc-oxide nodes. This could be traced back to an essentially linear increase in the interface thermal insulance of each charge-transport channel with linker length. Simulations on a suitably chosen model system revealed that this behavior is a direct consequence of the MOF framework structure and topology. For IRMOF-10 with biphenyl-based linkers, we additionally observed that the situation deteriorates further due to a strongly strained structure of nodes and linkers, which are caused by the atomistic details of the linker geometry.

Conversely, for rigid-backbone, pyrene-based linkers (IRMOF-14), the calculated thermal conductivity somewhat exceeded the original expectation. On the one hand, this can be associated with a more rigid linker reducing phonon scattering. On the other hand, for this MOF, the transport-channel insulance of the interface is also essentially the same as that for the much shorter phenylenedicarboxalyte linker. This suggests that the key factor determining the thermal resistance of a node-linker interface might not be the physical length of the used linker but rather the number of rigid objects it contains. Notably, for anisotropic MOFs the situation is further complicated by the observation that the cross-linkers perpendicular to the heat-transport direction also profoundly impact the transport-channel insulance of the interface and, thus, the thermal conductivity (beyond mere geometric effects). When combining, e.g., pyrene-based linkers in the heat transport direction with phenylene-based ones perpendicular to it, at least part of the beneficial impact of the rigidification of the backbone in pyrene is lost. A profound effect on the perpendicular linkers on thermal transport is also observed for the considered model systems.

These considerations (whose key points are summarized in Figure 1) show that structure-to-property relations for heat transport in MOFs arise from a complex interplay between MOF structure and topology with the details of the linker chemical structure and length, where it is additionally known from previous studies that the mass of the employed metal ions, as well as the strength between nodes and linkers, also play a crucial role [50]. Thus, one has to conclude that, at this stage, we are only at the beginning of a proper understanding of heat transport in this complex class of materials.

## Data Availability

The data presented in this study are openly available in the NOMAD Repository and Archive at https://www.doi.org/10.17172/NOMAD/2022.06.02-1, reference number dnyf3l38R7CcNaDfd-BDQA.

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
