# Peer review of "Exploring the Impact of the Linker Length on Heat Transport in Metal–Organic Frameworks"

_nanomaterials, 2022, doi:10.3390/nano12132142_

Round 1
Reviewer 1 Report
This work demonstrated the computational study regarding the effect of linker length on the thermal conductivity and heat transport bahaviors of IRMOF system. The comprehensive computational studies were done systematically. The obtained result for IRMOF-1 is also consistent with the reported value from experiments. The findings here are indeed new in literature and do provide new insights to the MOF community. Since the discussion has been made nicely to interpret the data and the data can fully support the conclusions, I would recommend the publication of this paper in Nanomaterials as it.
Author Response
We thank the reviewer for the very positive assessment of our manuscript.
Reviewer 2 Report
This manuscript by Prof. Zojer et al. describes the computational investigations of thermal conductivity of isoreticular MOFs. The author has taken a series of MOFs bearing di-carboxylate linkers with various length to evaluate the relationship between the thermal conductivity and the linker length. The node involvement is also considered. The study is interesting and could be of potential to the design of new MOFs for specific applications.
It is concluded that rigidity of the framework and pore sizes of the MOFs have a great influence on the calculated thermal conductivity. In Figure 4, the calculated thermal conductivity is plotted against delta Z(unit). The author might want to make a plot demonstrating the calculated thermal conductivity vs pore size, in which values are obtained experimentally.
It would be interesting to see the prediction on MOFs bearing NDI (naphthalene diimide), RDI, PMI moieties. Rigid, various length, various pore sizes, several ligand features for comparison.
Author Response
We thank the reviewer for the very positive assessment of our manuscript.
----
"It is concluded that rigidity of the framework and pore sizes of the MOFs have a great influence on the calculated thermal conductivity. In Figure 4, the calculated thermal conductivity is plotted against delta Z(unit). The author might want to make a plot demonstrating the calculated thermal conductivity vs pore size, in which values are obtained experimentally."
----
A plot of the calculated thermal conductivities of the studied model systems as a function of the inverse area of the unit cell perpendicular to the heat transport direction have been added to the Supplementary Materials as Figure S32(c). Additionally, Figure S14 (where the pore size dependence of the thermal conductivity is shown for the actual MOFs) is now clearly referenced in the main manuscript. As far as experimental values are concerned, of the MOFs modelled in the present study IRMOF-1 is the only system for which experimental data exist. The corresponding value has now been added as a data point in Figure 4.
As far as other MOFs are concerned, the number of studies in literature is rather limited. To the best of our knowledge, values exist ZIF-8, MOF-5, CU-BTC, ZIF-4, ZIF-62, UiO-66, UiO-67, MIL-101 (Cr), a perovskite-type-IRMOF, Ni3(HITP)2), µp-AF. The pore size of most of these systems is, however, rather similar and to make things worse these systems are vastly different with respect to topologies and coordination environments. Consequently, the thermal conductivities of these systems are not really comparable.
In fact, as shown for ZIF variants in a computational study in doi:10.1021/acsami.0c21220, changes in heat transport pathways can cause a strong deviation from the linear trend of the thermal conductivity with the inverse pore cross-sectional area (which we mention in the introduction). To cut a long story short, no experimental study has been carried out so far, that measures the thermal conductivity in MOFs with systematically increased linker lengths with a consistent sample and measurement setup. For example, MOFs have been studied as pellets, as single crystal, as thin films, etc.
Therefore, we refrained from compiling a plot comparing different experimental values.
----
It would be interesting to see the prediction on MOFs bearing NDI (naphthalene diimide), RDI, PMI moieties. Rigid, various length, various pore sizes, several ligand features for comparison.
----
We agree to the reviewer that including additional systems into our comparison would be highly desirable. The development of a force field of satisfactory quality, however, took us usually several weeks and sometimes several months for each system. Therefore, the study of additional systems goes beyond the scope of the present manuscript. In this context it is worthwhile mentioning that we are currently exploring the potential of novel machine-learned force fields that appear to be of exceptional quality (at least for calculating phonon properties) and where the parametrization can be done in an automated manner. We are currently evaluating their suitability for studying thermal transport and in case this assessment is positive, we ought to be able to achieve a much higher throughput in thermal conductivity calculations in the future.
Reviewer 3 Report
In the paper entitled “Exploring the impact of the linker length on heat transport in metal-organic frameworks" the Authors systematically studied the impact of the linker length on heat-transport in MOFs by means of molecular-dynamics simulations. MOFs' properties modeling is a very interesting research field and papers dealing with such a topic are highly welcome. The manuscript is well organized and nicely written and a lot of additional information are provided as supporting material. The figures are of good quality. I have no specific comments or criticisms about the technical content of the manuscript. The manuscript is suitable for publication to me after minor revisions. I think that a further improvement of the introduction section to give a complete overview of the state of the art regarding modeling studies on MOF heat transfer should be performed. In addition, I suggest to propose a scheme to resume the main findings of the study, in particular, I suggest to make a “Graphical Conclusions’ scheme" reporting the effects of the structural parameters on the heat transfer as established by modeling studies.
Author Response
We thank the reviewer for the very positive assessment of our manuscript.
----
"I think that a further improvement of the introduction section to give a complete overview of the state of the art regarding modeling studies on MOF heat transfer should be performed."
----
We have extensively expanded the introduction providing a rather comprehensive overview of the main conclusions of theoretical and experimental studies dealing with the investigation of various aspects of thermal transport in MOFs on page 2 (adding also 15 references).
----
In addition, I suggest to propose a scheme to resume the main findings of the study, in particular, I suggest to make a “Graphical Conclusions’ scheme" reporting the effects of the structural parameters on the heat transfer as established by modeling studies.
----
We have added such a scheme in the conclusion section.